# A Novel Location Privacy-Preserving Approach Based on Blockchain

**DOI:** 10.3390/s20123519

**Published:** 2020-06-21

**Authors:** Ying Qiu, Yi Liu, Xuan Li, Jiahui Chen

**Affiliations:** School of Computer, Guangdong University of Technology, Guangzhou 510006, China; qiuying@mail2.gdut.edu.cn (Y.Q.); lx@mail2.gdut.edu.cn (X.L.); csjhchen@gdut.edu.cn (J.C.)

**Keywords:** location-based services, location privacy-preserving, blockchain, k-anonymity

## Abstract

Location-based services (LBS) bring convenience to people’s lives but are also accompanied with privacy leakages. To protect the privacy of LBS users, many location privacy protection algorithms were proposed. However, these algorithms often have difficulty to maintain a balance between service quality and user privacy. In this paper, we first overview the shortcomings of the existing two privacy protection architectures and privacy protection technologies, then we propose a location privacy protection method based on blockchain. Our method satisfies the principle of k-anonymity privacy protection and does not need the help of trusted third-party anonymizing servers. The combination of multiple private blockchains can disperse the user’s transaction records, which can provide users with stronger location privacy protection and will not reduce the quality of service. We also propose a reward mechanism to encourage user participation. Finally, we implement our approach in the Remix blockchain to show the efficiency, which further indicates the potential application prospect for the distributed network environment.

## 1. Introduction

With the rapid development of communication technology, location-based services (LBS) are being widely used in various fields [1,2,3]—for example, health care, mobile social. LBS [4,5] is based on the location information, with the support of geographic information systems (GIS) and lightweight mobile devices, to provide users with value-added services including points of interest query. LBS is divided into snapshot query and continuous query [6]. The snapshot query refers to the user actively input query conditions to query, such as “query the gas stations closest to me now.” The continuous query means that the location service provider (LSP) provides location services according to the continuous changes of the user’s location, such as “query the gas stations closest to me while driving.” Currently, many applications based on location services have been developed. Typical applications include map applications (such as Google Maps), interest point query (such as Meituan), location-aware services (such as Foursquare), etc. [7]. Users need to share their location data when using these applications. For example, when users want to “query which Meituan takeaways are near me,” users need to provide location data to the LSP. The more location information users share, the better location services they maybe obtain from LBS. However, the attacker’s inference attack on the location data can analyze users’ sensitive information, such as personal data, workplace, and health status [8,9]. Therefore, how to balance location services from LBS and users’ privacy leakage has become an urgent problem [10,11,12]. 

Existing location privacy protection architectures can be roughly categorized into decentralized architecture and centralized architecture [13]. The centralized architecture introduces a fully trusted third party (TTP) anonymizing server [14]. The anonymizing servers receive the precise location and query content (as shown in Figure 1) and then use spatial obfuscation, location perturbations, pseudonym, and other technologies to protect the user’s location privacy [15,16]. On the one side, the anonymizing servers have all information of the user, and it will become a bottleneck that bring security threats once it has been attacked. On the other hand, privacy protection technologies, such as spatial obfuscation and location perturbations, are difficult to balance with user privacy and location service quality. The pseudonym technology must rely on a trusted third-party server. Finally, heavy computing tasks can easily make anonymizing servers the performance bottleneck of the structure. 

Contrary to the centralized architecture, the decentralized architecture does not require TTP anonymizing servers. In the decentralized architecture, the client communicates directly with the LSP and uses encryption and user collaborative technologies on the client to protect user privacy. However, these privacy protection technologies still have some flaws. For example, encryption-based technology requires the client to have strong computing processing capabilities. Continuous encryption and decryption operations in continuous query will also seriously affect the quality of location services. User collaborative-based technologies cannot screen out malicious or even offensive collaborative users. Thus, there is the potential threat of privacy leakage. Sometimes it is even impossible to successfully construct an anonymous area due to the lack of collaborative users. To sum up, our goal is to improve the deficiencies of the existing privacy protection architectures and privacy protection technologies to make it more effective in balancing the quality of location queries and user privacy. 

A novel shared ledger-blockchain is attracting great attention from researchers. In the blockchain network, users use virtual cryptocurrencies [17,18] for transactions and use asymmetric cryptographic algorithm to protect their true identity (*ID*). In addition, blockchain technology does not depend on third parties and achieves decentralization of the system through consensus mechanism. The current blockchain system can be divided into two categories—permissioned and public [19]. Permissioned blockchain can be divided into private blockchain and consortium blockchain. In the public blockchain, everyone can join the consensus process and get services. So, it will take plenty of time to propagate transactions and blocks. Different from public blockchain, everyone needs to be certificated to join the consensus process in private blockchain. Transactions on private blockchains are fast and efficient, but a single private blockchain network cannot effectively protect user privacy in location services [20]. 

The *k*-anonymity mechanism is the most common method for location privacy protection. It hides the user’s real location in *k*-1 false locations [21,22]. Compared with other methods, *k*-anonymity does not rely on complex cryptographic techniques. This can effectively reduce the user’s computation overhead and allow users to enjoy a higher location service quality. 

Based on structural characteristics of the blockchain and the advantages of the *k*-anonymity, we will combine *k*-anonymity and multiple private blockchain networks to solve the problems above. In this article, we propose a location privacy protection method based on blockchain. The major technical contributions of this paper are as follows:We propose a novel decentralized location privacy protected architecture to protect user’s location privacy in LBS. The framework uses multiple private blockchain networks to decentralize user transaction records, thereby enabling the system to achieve decentralization.We use nodes on the private blockchain to replace the user and send query requests to the LSP, then we return the query result provided by the LSP. The method cuts off the direct contact between the LSP and the user and further enhances the user’s location privacy protection.We use the *k*-anonymity principle to obfuscate the user’s actual location. The framework does not require complicated cryptographic techniques and algorithms. It enhances the location privacy of users and can also get the most accurate location services.We propose a reward distribution mechanism to incentivize user participation, and we use the characteristics of blockchain smart contracts to ensure that transactions are fair and enforceable, giving users better safeguard.We analyze the efficiency, privacy and security of the proposed system through a series of simulation experiments.

The rest of the paper is organized as follows. The related work is summarized in Section 2. Preliminaries are discussed in Section 3. In Section 4, we present the proposed system framework and attack model. We outline the location privacy protection scheme in Section 5. In Section 6, we present the scheme analysis. In Section 7, we detail the results of the performance evaluation, and in Section 8, we conclude the paper.

## 2. Related Work

In this section, we review existing research on centralized architecture and decentralized architecture for the application of blockchain in LBS.

### 2.1. Existing Research On Two Architectures

**The centralized architecture.** In the direction of spatial obfuscation, Gruteser et al. [23] first used the *k*-anonymity mechanism to obfuscate the user’s location in the anonymizing servers. These methods of generalizing the user’s location into a larger area or adding more other locations [24] will reduce the accuracy of the location query, resulting in a reduction in the quality of location services. Similar to spatial obfuscation, location perturbations will also reduce the quality of location services. For example, Andrés et al. [25] achieved a geo-indistinguishability mechanism. The algorithm uses controlled random noise to confuse the user’s exact location. Yin et al. [26] proposed a method to satisfy the differential privacy mechanism, using the Laplace mechanism to add noise. Although these methods consider the quality of location services, they still cannot fully balance the privacy of users and the quality of location services. The pseudonym approach is like an anonymous property of blockchain technology. This method of changing the user’s ID in anonymizing servers to hide their true identity [27] must find a reliable third party. 

Some scholars have also proposed ways to combine different privacy protection technologies. For example, Zhang et al. [28] proposed a method combining *k*-anonymity and order preserving symmetric encryption (OPSE) technology. The users first need to find *k*-1 reliable users around them. Although the third-party performance bottleneck of the centralized architecture is improved, this solution still requires semi-TTP anonymizing servers. Han et al. [29] used multi-server architecture to cut off the direct connection between the LSP and the user and used differential privacy mechanism to enhance privacy protection. Although this privacy protection architecture can obtain good service quality, social network resources cannot be completely trusted. In addition, the proposed differential privacy mechanism will also reduce the service quality.

**The decentralized architecture.** Distributed *k*-anonymity is a typical representative. For example, Chow et al. [30] used the point-to-point communication hop count method to obtain the location information provided by cooperative users, which will increase the network transmission delay. In order to solve this problem, Chow et al. [31] again used the actual location of historical collaboration users to construct anonymous zones. Peng et al. [32] obtained the actual location of collaborative users by sending a false collaboration request and store them in the cache. In the next LBS query, the anonymous area can be constructed using the location information in the cache. However, these methods require users to provide a large storage space. Hwang et al. [33] proposed a method that users can use the internet to obtain the actual location of construct an anonymous area. The above is to construct an anonymous area by finding collaborative users. Collaborating users may be malicious, so the privacy of users cannot be fully guaranteed. 

Other scholars have proposed dummy-based methods to construct anonymous zones [34,35,36]. These algorithms generate virtual users on the client and use the virtual user’s location to construct an anonymous zone. However, they will be constrained by the actual environment. To solve this problem, Hara et al. [37] and Suzuki et al. [38] proposed a method that hide the user’s location on the mobile terminal by generating dummies around the user. However, assumptions (user always stays in motion) related to users of these algorithms are practically untenable.

In addition, there are some encryption-based privacy protection technologies. For example, Yi et al. [39] used homomorphic encryption technology to achieve simultaneous protection of user location and query privacy. Encryption technology requires the client to have strong calculation processing performance, and usually cannot effectively balance privacy protection and location service quality.

### 2.2. Application Of Blockchain

Thanks to the beneficial characteristics, blockchain technology is widely used by the researchers in LBS. But only a small part is to protect location privacy in LBS. For example, Jia et al. [40] proposed a method based on privacy protection to encourage users to actively participate in location services in intelligence crowd sensing networks. This method uses the tamper-proof property of the blockchain. Its main purpose is to motivate users to participate in location services. In location-based services, the authenticity of user location information is also a very important issue. In response to the defects of centralized verification approaches proposed by scholars, Amoretti et al. [41] proposed a proof-of-location scheme based on blockchain technology. This solution is used for LBS to verify the existence of a user’s specific geographic location. In the vehicular ad-hoc network (VANET), Luo et al. [42] proposed to record the credit of the vehicle on the blockchain to assist the vehicle to avoid malicious tracking by other participants. The main purpose of these methods is not to protect location privacy in LBS. To this end, Yang et al. [43] proposed a privacy-preservation crowdsensing system based on blockchain for the defects of the centralized structure in the crowd perception system and the leakage of user location privacy. This system combines public blockchain network and multiple private blockchain networks to distribution user transaction records and protect user location privacy. However, it only discusses the application of blockchain in crowd sensing networks.

Based on the inspiration of Han et al. [29] and the above discussion, we propose a new location privacy protection method to explore *k*-anonymity privacy protection from the perspective of blockchain.

## 3. Preliminaries

### 3.1. Blockchain

In 2008, Satoshi Nakamoto invented the blockchain [44]. Blockchain is a new application technology combining distributed storage, consensus mechanism, asymmetric cryptographic algorithm, smart contract, and other technologies [45,46]. The shared ledger records all user transactions in the blockchain network. The consensus mechanism means that all nodes in the blockchain need to reach consensus when recording transactions. It effectively prevents data from being tampered with. The blockchain uses an asymmetric cryptographic algorithm to anonymously process the user’s identity. The public key is used as the user’s account address, so the user’s real identity is difficult to be discovered by others. The smart contract is a micro executable program that can automatically run after triggering certain conditions in the blockchain. It allows trusted transactions and agreements to be implemented between any P2P network.

Blockchain nodes are managed by all computer equipment (such as mobile phones, servers, etc.) participating in the blockchain. In this article, we collectively refer to these devices as blockchain nodes. Blockchain nodes are connected to each other in the network.

### 3.2. k-Anonymity

*k*-anonymity supplies data privacy with low accuracy by generalizing and hiding some attributes [21]. The larger the *k*, the lower the availability of data is, but the higher the degree of user privacy protection. In location privacy protection, *k*-anonymity hides the user’s exact location in *k*-1 dummy locations, keeping the query content unchanged. The larger *k*, the lower the user’s risk of privacy leakage, but the quality of the service obtained will decline.

## 4. Notations Definition and System Model

### 4.1. Notations

Let *U*= {*U_1_*, *U_2_*, *U_3_*, …, *U_n_*} be the set of requesters, and *A*= {*a_1_*, *a_2_*, *a_3_*, …, *a_k_*} be the set of rewards for the agent to upload a task. *T*= {*T_1_*, *T_2_*, …, *T_k_*} be the set of tasks, that is, *k* query requests. *t* is set by the requester, means the longest time it takes for the agent to complete *T_k_*. *T_r_* is set by the requester, means the longest time it takes for the agent to complete *T*. (*X_i_*, *Y_i_*) is the location of the query request. *q_c_* is the query content. *q_r_* is the query result. More notations are shown in Table 1.

### 4.2. System Model

The framework of our location privacy protection method is shown in Figure 2.

The following three parties are involved in the proposed framework:
(1)The requester is the user who needs to get location services. The requester releases tasks to the private blockchain to initiate a transaction, and then use the smart contract in the private blockchain to obtain sufficiently location services. The creator of the private blockchain and other nodes in the private blockchain can become requester.(2)The agent takes the place of the requester to send query requests to the LSP and return the query results to the user. They can choose to undertake tasks from the private blockchain based on their own privacy considerations. According to the reward distribution mechanism, the first node that accomplishes the task gets the most rewards, the last node that accomplishes the task gets the lowest rewards.(3)The miner is responsible for verifying the service results uploaded by the agent and recording the new transaction in the distributed ledger. Miners can get transaction fees and rewards if they successfully record new transactions. In addition, a miner can also be requester that releases a task or agent that undertakes a task.In the proposed system, the requester releases the task to the private blockchain. The agent undertakes the task from the private blockchain and completes the task within a specified time in exchange for rewards.

### 4.3. Attacker and Attack Strategy

We assume that all participants in the blockchain network are untrusted. The attacker can be any participating node in the blockchain. Below we summarize attack strategies and three kinds of roles that may threaten our system. They satisfy Honest-But-Curious (HBC) adversary model [47].

Attacker:
(1)Creator of private blockchain: Although the creator is also the requester, he/she can also be an attacker. The creator can save the transaction records of blockchain nodes on his own network.(2)Agent: The agent can obtain the transaction records of other requesters on the participating private blockchain and stores the transaction records it has downloaded on the network. However, due to the membership control protocol of the private blockchain, the agent cannot join all private blockchain networks. So, the attacker cannot fully obtain all transaction records of the same requester.(3)Agent colludes with creator: the malicious agent obtains the benefit by providing the user’s location information to the creator.Attack strategy: The attacker analyzes the user’s actual location by tracking the transaction records of the same account.

## 5. Our Location Privacy Protection Method

In this section, we introduce the proposed decentralized location privacy protection method based on blockchain. The proposed scheme reduces the risk of location privacy leakage during the process of sending query requests and receiving query service results.

### 5.1. Overview

Using blockchain’s traditional advantages in anonymization, independence, and decentralization, we propose a location privacy protection scheme. The blockchain network replaces third-party anonymizing servers and can overcome the weaknesses of third-party servers. Specifically, users can choose to create any number of private blockchains or participate in other private blockchains to issue query requests to distribute their transaction records. The nodes on the private blockchain can download the query request issued by the user and then sends the query request to the LSP. Finally, the nodes return the query result provided by the LSP to the user to get a reward. These nodes cut off the direct connection between the LSP and the user. It is hard for an attacker to collect all transaction records to infer the true *ID* of the user. Even if the attackers have some background knowledge, it is difficult to extract the user’s sensitive information. The user’s actual query request is hidden in the query content sent by the node, so the LBS server can provide the most accurate location services while ensuring that privacy.

The general process of the proposed blockchain based system is shown in Figure 3.

Create or join a private blockchain: The users can create their own private blockchain network, or they can choose to join private blockchain network created by other users. No matter they are creators or other nodes on the blockchain, they can release tasks as requesters.
Create private blockchain: When users want to obtain location services as a requester, they can create their own private blockchain. Then the user releases the task to the blockchain to initiate a transaction or become an agent to undertake the task released by other requesters on the blockchain.Join private blockchain: When users want to obtain location services as a requester, they can also apply to join others private blockchain release task to initiate a transaction or as an agent to undertake the task released by other requesters on the blockchain.Release query request: The requester releases a query request to the blockchain and sets the reward for the agent based on the resource consumption of the query request.Undertake query request: The agent downloads the task from the blockchain and uploads the query results to the blockchain within a specified time. Eligible data will be accepted and recorded, and corresponding agents will be rewarded. If the data is found to be unqualified, the agent will lose deposit.Reward distribution: If the data uploaded within the specified time is qualified, the smart contract will automatically distribute the reward to the agent.

### 5.2. Implementation Of The Proposed System

To ensure fair transactions in the above process, the requester creates a smart contract. Figure 4 shows components of the smart contract.

Requester *ID* indicates the owner of the task. Agent *ID* indicates the node that has accepted the task. Rewards show the cryptocurrency that the agent receives after completion. The requester needs to pay a deposit, which is automatically withdrawn by the smart contract. Agents also need to pay a deposit, which is used to prevent a malicious agent from performing a task but refusing to submit appropriate data. The agent needs to submit service results. The evaluation function is used to verify whether the location service results are appropriate, and the miner verifies whether the results are qualified.

#### 5.2.1. Create or Join Private Blockchain

Create private blockchain: The user can create a private blockchain by him/herself and then release query requests on the created private blockchains or become agent to undertake the tasks released by other requesters on the blockchain.Join private blockchain: The user can join as many private blockchains as he/she needs. The user must be verified by a set of rules established by the private blockchain owner in the private blockchain. In the proposed system, smart contracts verify that nodes are eligible to join the network by checking the following two conditions:
Is the number of nodes below the private blockchain research limit?Has the account been cleared three times in the private blockchain? (If the agent is found to download tasks maliciously but does not upload the service results, the agent will be cleared from the blockchain.)

Specifically, users submit their requests to the private blockchain creator to indicate that they are interested in joining the private blockchain network. If the user is authenticated, the request should be allowable. Algorithm 1 shows the details of the authentication process. Bulleted lists look like this:
**Algorithm 1.** Authentication process in joining private blockchain**Require:***U_type_***Ensure:***pk*, *sk*, *U_id_*, permit1: permit = false;2: {*pk*, *sk*} ←KeyGenerator ();3: ID_u_ ←pk;4: *U_type_* ∈ {Requester, agent};5: **if**
*ID_u_ been cleared three times or Blockchain is full*
**then**6:   **return** permit7:  **else if**
*ID_u_*
*∈ Pool_u_*
**then**8:    permit = true;9:  **else if**
*Pool_u_ ←Pool_u_ ∪ {ID_u_}*
**then**  permit = true;10: **end if**11: **return** permit

As is shown in Algorithm 1, the variable permit is an indicator of whether authentication was passed. In step 2, a pair of secret keys is generated. In step 3, the public key is used as the user’s *ID*. *U_type_* indicates the type of the registered user, which could be a requester or an agent. Steps 5 and 6 show that if the user account *ID* has been cleared three times on the chain or the node has reached the upper limit, the verification fails. If both conditions are not met, the algorithm skips to step 7 and verify whether the account *ID* is in the user pool. If so, the verification passes. Otherwise, the algorithm assigns a pair of keys to the user and records the keys in the user pool (steps 8 and 9).

#### 5.2.2. Release Query Request

The requester releases *k* query requests to the blockchain and needs to deposit a certain amount of cryptocurrency into the blockchain in advance. The smart contract will deduct enough cryptocurrency to allocate to the agent according to the reward amount set by the requester. 

Initialization tasks: The requester submits *k* values, *R* values, and the transaction *T_r_* to the system. The smart contract deducts enough cryptocurrency from the deposit of the requester based on the values of *k* and *R*.Construct query request: Requester construct k query requests on the mobile terminal, that is, the task set *T* {*T_1_*, *T_2_*, …, *T_k_*}, *T_i_*= [(*X_i_,Y_i_*), *q_c_*]. To reduce user resource consumption, the range of *k* is set to 5 < *k* < 10. The user enters *k* tasks on the mobile terminal.Release query request: The requester releases a task to the blockchain to initiate a transaction. The format of the task is <*T_i_*, *R*, *t*>. The agent can select tasks based on how much *R* is. Therefore, the success of the task also depends on the remuneration that the requester is willing to pay the agent.

#### 5.2.3. Undertake Query Request

After the agent joins the blockchain, he needs to deposit a certain amount of cryptocurrency into the blockchain in advance. The requester releases the task to the blockchain. The agent chooses a task and downloads the query request according to the reward (*R*) by the requester, the specified time (*t*) by the requester, and his own privacy. If the agent downloads the task and does not submit the result within a specified time, the task will be forcibly withdrawn by requester and the deposit is deducted as a penalty. 

Details of the task publishing and downloading process are shown in Algorithm 2.
**Algorithm 2.** The process of task release and download**Require:***k* query requests, *R*, *t***Ensure:** Qualified service results1: procedure = end;2: {*k* values, *R* values, *T_r_*} ← requester;3: k query requests ← requester;4: blockchain ← < *T_i_*, *R*, *t*>;5: agent download task;6: the LSP ← the query request, the query result ← the LSP;7. **while**
*task set running time* <= *T_r_*8.   **if**
*all tasks in the task set are submitted and verified successfully*
**then**9.     **break**10.   **else if**
*service result is submitted and qualified*
**then**11.     the single task ends; 12.    **else** the requester forcibly withdraws the task;13.   **end if**13. **end while**14. **return** procedure

In Algorithm 2, the variable procedure is an indicator that task end. In step 2, the requester submits the value of *k*, the value of *R*, and the value of *T_r_* to the system. The system deducts enough cryptocurrency from the deposit of the requester according to the values of *k* and *R*. In step 3, the requester constructs *k* query requests on the mobile terminal. Step 4 shows that the server publishes *k* query requests to the blockchain in the form of <*T_i_*, *R*, *t*>. The agent views the single task <*T_i_*, *R*, *t*> in the blockchain network, and then downloads it (step 5). After downloading the task, the agent privately obtains the query results from the LSP (step 6). When the task set running time does not exceed *T_r_* (step 7), perform the following steps. If all task results in the task set are submitted and verified successfully (step 8), the task is ended early (step 9). Otherwise, the algorithm skips to step 10 to verify whether the agent has submitted the service result and is qualified. If so, the single task end and the agent can get a reward (step 11). Otherwise, the requester forcibly withdraws the task, the agent deposit is deducted, and the task can be downloaded again by other agents (step 12).

#### 5.2.4. Reward Distribution

If the agent submits the query results within the specified time, it can get a certain amount of reward. The first agent who submits the query result will receive the highest reward, to motivate the agent to submit the service result in time after downloading the task. According to the experiment, the resource consumed by the agent from downloading the task to submit the service result is small.

When initializing tasks, the requester first submits an *R* to the system. In this process, we use the rules of arithmetic progression [48] to set rewards. The smart contract deducts enough cryptocurrency *R_s_* from the deposit according to the summation formula of the arithmetic progression. Let the common difference be *d* = *R*, *a_1_* = *R*. *R_s_* = [(*R* * *k* + *R*) * *k*]/2, where *k* is the current *k* value submitted by the user. But when the task fails, *Rs* are allocated to the agents (*k_f_*) actually submitting the task according to the reward differential distribution mechanism. *k_f_* is counted by how many results is returned. For example, if the task is failed, the requester does not obtain *k* results within the specified time. The reward for the last agent to submit a task is *R_f_*=2*R_s_*/(*k_f_*^2+*k_f_*).

Details of reward distribution are shown in Algorithm 3.
**Algorithm 3**. Reward distribution**Require:***k**_f_*, *R***Ensure:***R_s_*, *R_f_*1: Requester submits currency value *R*;2: Smart contract deduction *R_s_*;3: **If**
*Task success*
**then**4:  **return**
*K* agents get rewards5: **else**
*k**_f_* agents get rewards6: **end if**

In Algorithm 3, the system has deducted enough cryptocurrencies (*R_s_*) from the deposit of the requester, that is, Steps 1 and 2 in Algorithm 3. As is shown in Algorithm 3, if the task is successful, the agents will get rewards (step 4). The reward for the last agent to submit a task is *R*. Increasing by *d* = *R*, the reward for the first agent submitting the task is *k* * *R*. If the task is unsuccessful, then distribute *R_s_* to the agent who actually submitted the task (*k_f_* < *k*, step 5). The reward for the last agent to submit a task is *R_f_*. Increasing by *d* = *R_f_*, the reward for the first agent submitting the task is *k_f_* * *R_f_*. In a blockchain network, the smart contract will automatically pay the user in encrypted currency through the user’s public key. In this process, the smart contract does not know the true identity of the user.

## 6. Scheme Analysis 

### 6.1. Privacy Analysis

We will analyze the privacy of the proposed scheme through the blockchain structure and *k*-anonymity in this section.

The location privacy protection architecture based on the private blockchain is benefited from the structural characteristics of the blockchain, eliminating the technical bottleneck of third-party anonymizing servers. Users can decentralize transaction records through multiple private blockchain networks and can also handle multiple accounts to prevent attackers from tracing their true identity. Even if the private blockchain creator is an attacker, he cannot possibly collude with other blockchain owner. The blockchain can effectively prevent data from being tampered with and stolen. In the proposed solution, users have been cleared three times on the same blockchain network, which effectively prevents attackers from lurking on the chain to collect other users’ transaction records. We utilize the participants on the blockchain to obtain services, cut off the direct contact between users and LSPs. Based on the anonymous property of the blockchain, it is difficult for LSPs to track users who really need to obtain location services. 

The *k*-anonymity principle can effectively protect users from background knowledge attacks. If attackers launch an identity attack on data-based query reasoning during the data release process, *k*-anonymity can effectively prevent users from leaking sensitive data in the process. Generally, the larger the value of *k* is selected, the higher the degree of privacy. However, it also means that the lower the data availability, the worse the service quality. In the proposed scheme, we set the range of k be 5 < *k* < 10. The user directly enters *k* query requests in the mobile terminal, which includes the user’s actual location and *k*-1 false locations. When *k* < 5, the degree of anonymity is too low, and the user’s location privacy cannot be effectively protected while ensuring service quality; when *k* > 10, the user needs to enter more than 10 locations on the mobile terminal, which may increase the user’s query overhead in the subsequent query process. Therefore, in the proposed scheme, the setting of the *k* value can obtain the most accurate location service while ensuring the high degree of privacy protection. 

### 6.2. Security Analysis

Three possible attackers and attack strategy in our system are discussed in Section 4.3. In fact, these threats can be collectively called HBC adversary model. Now, we will briefly discuss the impact of this adversary model on our proposed system.

The attackers follow the legal agreement in the system and explore the user’s privacy by collecting or tracking the user’s public information. In our scheme, the users’ transaction records will be scattered in many different private blockchain networks. The users can generate many key pairs at the same time to use different accounts to trade in the blockchain network. Even if the attacker gathers all the transaction records, it is difficult to analyze the user’s real identity. 

## 7. Performance Analysis

In order to illustrate and verify the proposed method, we use remix [49] to write smart contracts on the Ubuntu system, and the solidity compiled version is 0.5.1. We use the tool web3.js to interact with smart contracts, while the users will release query requests and obtain service results on web pages. We performed simulation experiments on the above platform. Then, we will analyze the performance of the proposed system from the following aspects: the efficiency of the blockchain, the success rate, the response time, the efficiency of the reward distribution mechanism. At the end, the proposed scheme is compared with the following: existing privacy protection architectures and a baseline method in Han et al. [29].

### 7.1. Efficiency Of The Blockchain

We analyze the efficiency of the systems of using the blockchain and not using the blockchain, from which we summarize the characteristics as follows:
Decentralization: The blockchain does not need to use an intermediate server, which can save a lot of server overhead.Anonymity: The blockchain uses cryptocurrency for transactions. The identity information of each node does not need to be disclosed or verified, and information transfer can be performed anonymously. Because blockchain technology allows people to collaborate on a large scale without the need of mutual trust, this also solves the untrustworthy vulnerability of collaborative users in user-collaboration technology.Smart contracts: The smart contracts can ensure the safety and fairness of transactions in the blockchain. This has a positive effect on transactions between users in the proposed system.Consensus mechanism: Every transaction on the blockchain network must be verified by a consensus mechanism, and each block will record a transaction timestamp [50]. Users can easily verify and view historical transactions through the access block.

However, the creation of a new block in the blockchain network has a very strict verification process [44], which will result in a delay in confirmation time. As a result, the application efficiency of using blockchain to obtain services is reduced. In our scheme, the requester calculates the reward amount privately, the agent obtains location services privately. Therefore, there is no heavy calculation task on the blockchain. Efficiency of the proposed system is acceptable.

### 7.2. Success Rate and Response Time

The response time is the time that requester uses the proposed system to obtain location services. The process of obtaining location services includes initializing tasks, requester to construct *k* query requests, agents downloading tasks, agents obtaining location services privately and uploading service results, and requesters obtain service results. The time required for the user to construct *k* query requests on the mobile terminal depends on the requester’s own situation. The time required for a single task is set by the requester himself, so the maximum time for the agent to obtain the location service in private and upload the service results can be selected by the requester himself. In summary, in the course of the experiment, it is only necessary to record the time of initialization task, the time of agent download task, and the time that the requester obtains the service results. Through a series of simulation experiments, we calculated the task success rate and the average response time required by the user to use the proposed system. 

The response time is count in seconds. No matter what *k* is, we will fix *t* = 25s, *T_r_* = 120s. We fix the value of *k*, and randomly publish 10 task sets on the web page at the same time. Finally, we calculate the situation of the obtained service results.
The task was unsuccessful: The requester did not obtain *k* service results within the specified time.The task was successful: The requester obtained *k* service results within the specified time. The success of the task includes the following two situations:
(1)The task was successful1: The agent uploads the service results in accordance with the regulations;(2)The task was successful2: A few agents did not upload the service results in accordance with the regulations. The requester forcibly retracted the task, and the task was downloaded again by other agents.

Table 2 shows the situation of obtaining service results when *k* = 5. “None” means failed to get service result. It can be seen from the table that task set 6 has failed. The other nine task sets successfully obtained all service results within 120 s. Figure 5 summarizes the status of all tasks obtaining service results, that is, the task success rate. As shown in Figure 5, when *k* = 5, there are nine successful task sets in the released 10 task sets, two of which belong to the second case of task success. As can be seen from the table, the task success rate is close to 100%.

Figure 6 shows the average response time, for the two cases where the task was successful. The response time units in the figure are all in milliseconds. It can be seen from the table that with *k* increases, the average response time is no more than 5 s. It can be seen that even when an agent fails to upload the service results as required, the required response time will not cause a large loss to the user.

**Remark** **1.**
*In Figure 5, all parameters are fixed, we randomly publish 10 task sets on the web page at the same time. Combined with the response time shown in Figure 6, it can be seen that when multiple tasks are published in the system at the same time. The performance of the system does not fluctuate much. *


**Remark** **2.**
*Scalability performance in large-scale scenarios: As can be seen from Figure 6, when 10 task sets are published on the web page at the same time, the requester’s time to obtain tasks is not more than 5 s, which is completely acceptable. Of course, when a large number of users use it, the task set released at the same time will definitely be many times more than 10. In Section 7.1, we mentioned that the creation of a new block in the blockchain network has a very strict verification process, which will result in a delay in confirmation time. As a result, the application efficiency of using blockchain to obtain services is reduced. In our scheme, the requester calculates the reward amount privately, the agent obtains location services privately. Therefore, there is no heavy calculation task on the blockchain. Efficiency of the proposed system is acceptable. On the other hand, the transaction on the private blockchain is usually fast, and the blockchain has been really applied to real life, i.e., the arbitration blockchain. Therefore, we believe that in large-scale scenarios, the scalability of the system will not fluctuate too much.*


**Remark** **3.**
*In terms of continuous query, according to the results of our experiment, the time for the requester to obtain the service result can be calculated in seconds. For example, “query the gas stations closest to me while driving,” the requester can release tasks continuously to obtain service in a short time. So, our method is also available for continuous query, but it is more suitable for snapshot query.*


### 7.3. Efficiency Of The Reward Distribution Mechanism

In the case where *k* and the task set are unchanged, we divide the reward mechanism into an average distribution mechanism and a differential distribution mechanism. Response time is used to reflect the execution efficiency of the reward differential distribution mechanism. As can be seen from Figure 7, when *k* < 7, the response time required by the differential distribution mechanism is significantly lower than the response time required by the average distribution. Since the calculation amount required by the differential distribution mechanism is greater than the average distribution mechanism, when *k* > 7, the response time of the differential mechanism is greater than the response time of the average mechanism. However, we can see from the figure that when *k* > 7, the time difference does not exceed one second at most. The data in the following figure prove that, compared with the average distribution rewards, the differential distribution mechanism can motivate agents to download preferentially and complete tasks quickly.

### 7.4. Comparison Of Solutions 

At the end of the performance analysis, we summarize the differences between the two existing architectures and our proposed architecture in Table 3. In addition, our scheme was proposed based on the inspiration of Han et al. [29]. The social network resources mentioned in Han et al. [29] cannot guarantee full credibility, so it cannot prevent attacks by malicious participants. On the other hand, it also uses a differential privacy mechanism to enhance the privacy protection of users, which will also reduce the quality of location service. We use blockchain to obtain location services, even if agents are malicious, they cannot track user’s true identity. In addition, we use the principle of *k*-anonymity in our article to enhance location privacy protection, so that users can get more accurate location services. Our work explores *k*-anonymity privacy protection from a blockchain perspective, which is a better attempt.

## 8. Conclusions

In this paper, we proposed a novel location privacy protection architecture. We use multiple private blockchains to protect location privacy without the help of a third party, and then use the nodes on the private blockchain to cut off the direct contact between the user and the LSP. Untrusted LBS servers or any other adversary can no longer directly access the user’s location information. We use the *k*-anonymity principle to enhance the privacy protection of users’ locations while enabling users to obtain the most accurate location services. The differential distribution mechanism proposed in this paper also improves the efficiency of the system operation and the user experience. Smart contracts guarantee fairness and enforceability of transactions. Through theoretical analysis and a series of simulation experiments, we verified that the solution does not need to use any complex algorithms and can provide the most accurate location services whereas enhancing location privacy protection.

However, our method is more suitable for snapshot query. In future work, we will improve our scheme to make it more efficient when using a continuous query. In addition, we will also apply our method to practical applications.

## Figures and Tables

**Figure 1 sensors-20-03519-f001:**
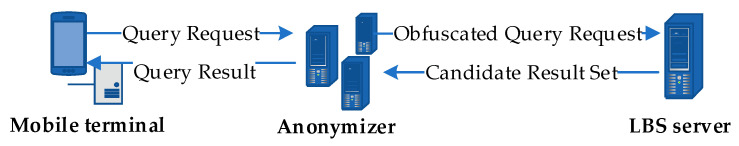
The centralized architecture.

**Figure 2 sensors-20-03519-f002:**
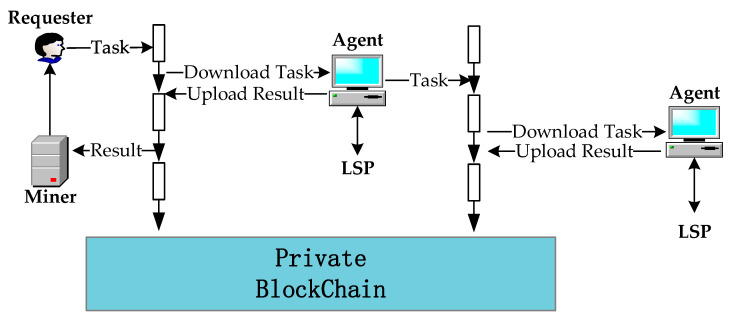
Framework of the proposed location privacy protection.

**Figure 3 sensors-20-03519-f003:**
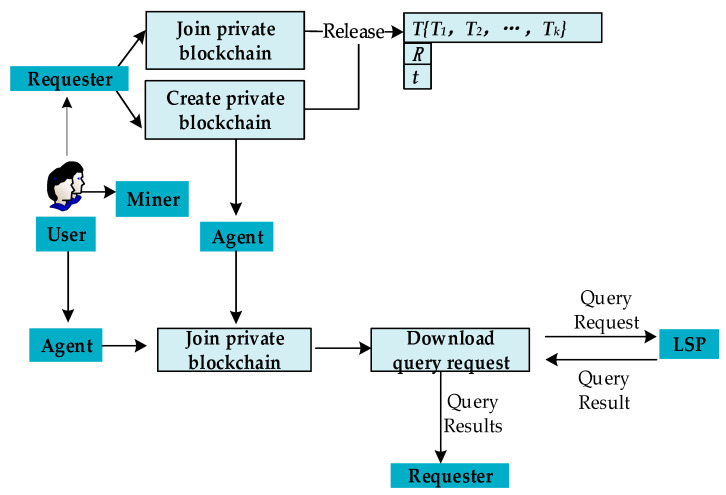
The general process of the proposed framework.

**Figure 4 sensors-20-03519-f004:**
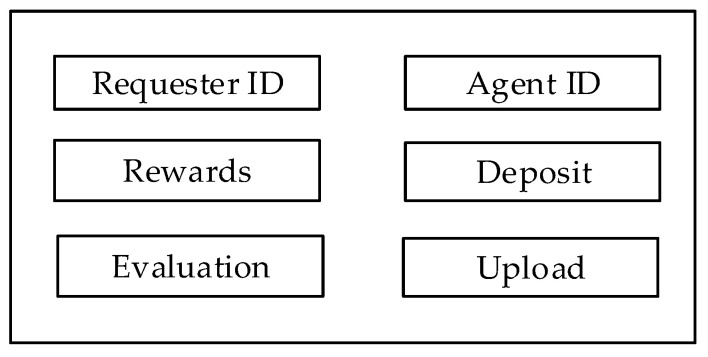
The components of smart contract.

**Figure 5 sensors-20-03519-f005:**
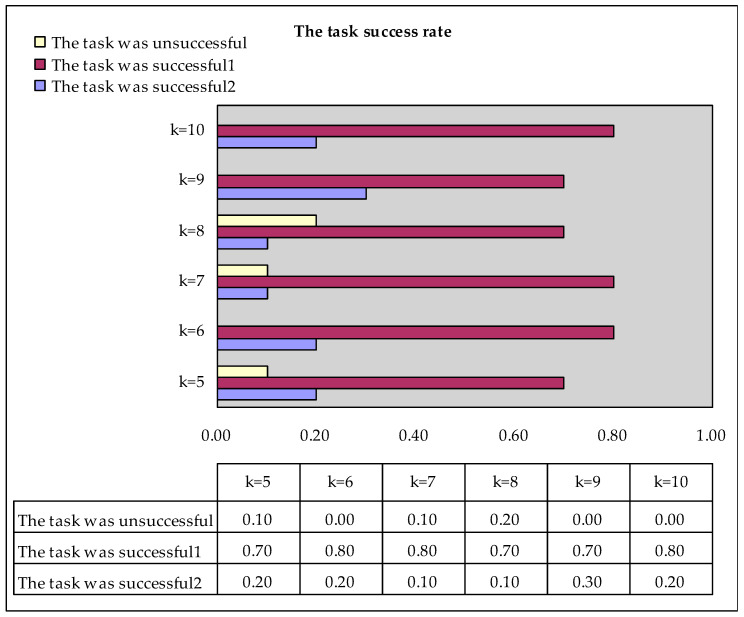
The task success rate.

**Figure 6 sensors-20-03519-f006:**
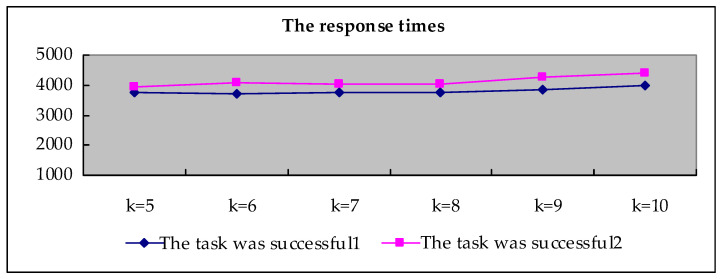
The response times.

**Figure 7 sensors-20-03519-f007:**
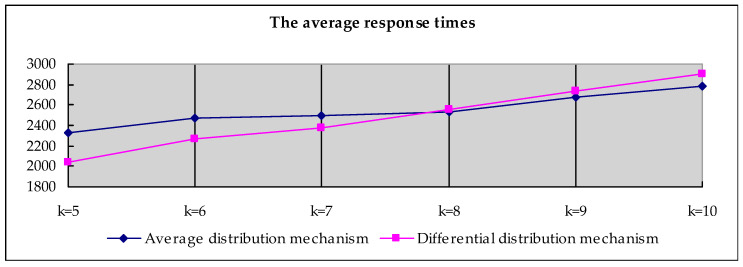
Efficiency comparison of two reward distribution mechanisms.

**Table 1 sensors-20-03519-t001:** Notations.

Notation	Description
*ID_u_*	The *ID* of the user (agent or requester)
*ID_r_*	The *ID* of the requester
*ID_a_*	The *ID* of the agent
*Pool_u_*	A set of user *ID*
*R*	Rewards for the last agent to upload a task when the task successful
*R_f_*	Rewards for the last agent to upload a task when the task fails
*R_s_*	Sum of rewards
*k_f_*	The final *k* when task fails

**Table 2 sensors-20-03519-t002:** The total time required to obtain service results when *k* = 5.

	Task Sets	1	2	3	4	5	6	7	8	9	10
*k* = 5											
*T_r_*		110s	95s	102s	89s	101s	None	100s	111s	114s	101s

**Table 3 sensors-20-03519-t003:** Comparison of methods.

Classification	Architectures	Privacy Protection Technologies	Service Quality and Privacy Protection	Computation Overhead
Our method	Decentralized	Blockchain and *k*-anonymity	Get good privacy protection while also getting high quality of service	Medium
Privacy protection architectures	Centralized	Spatial obfuscation [23,24]	The better the privacy protection, the lower the quality of location services	Medium
		Location perturbations [25,26]		
		Pseudonym [27]	Relying on third parties, third parties have become the biggest performance bottleneck	Low
	Decentralized	Encryption-based technology [39]	Usually can’t balance service quality and privacy protection well	High
		User collaborative-based technology [32]	The better the privacy protection, the lower the quality of location services	Medium
Paper [29]	Multi-server	Using social network resources to satisfy the principle of *k*-anonymity, differential privacy		Medium

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
