# Peer review of "A Novel Location Privacy-Preserving Approach Based on Blockchain"

_sensors, 2020, doi:10.3390/s20123519_

Round 1

Reviewer 1 Report

I believe that some of previous remarks still apply for this second version:

  • The contribution is not significant as well. The authors simply apply existing techniques from the blockchain  field for preserving a location privacy.
  • No real experimentation is done.  The proposed approach is validated by simulation.

Regarding, the cover letter:

  • The "I" pronoun is used instead of "We" even though the article has three authors. So this makes things not enough clear about who made the main contribution in this work. I fear that the contribution of the other authors is minimal.
  • Remark 3: the related work section is still missing a discussion paragraph which underlines the importance of the proposed approach with respect to existing works.
  • Remark 4: unfortunately the proposed answer in neither clear and nor convincing.
  • Remark 5: Why not keeping only one agent in this case.
  • Remark 7: One of our previous remarks is highlighted in yellow but no answer is proposed for it.
  • Remark 8: The authors say "the efficiency of our method
    in practical applications is not very clear". This is a negative position from the authors themselves which confirm our initial feeling about the immaturity of the proposed approach.
  • Remark 10: the authors say "There are indeed many low-level errors in the article". Once again the authors admit that the paper still contains errors and by the way is not ready for publication.

Reviewer 2 Report

Thanks to the authors for their effort to improve the paper. Many problems are resolved and I think it is possible to improve it further.

  1. The paper still has fundamental writing problems. For instance, some parts of it are written in first person form. “ Users are required to share their location data when using these applications. For example, 34 when I want to "query which Meituan takeaways are near me", I need to provide my location data to 35 the LSP. The more location information users share, the better location services they maybe obtain 36 from LBS. However, the attacker's inference attack on the location data can analyze user's sensitive 37 information such as the user's personal data, work place, and health status”
  2. I am still not understanding why do we need to create blocks. I would like to see 1-2 solid examples and scenarios which describe the contrast of having to create the blocks and not having the blocks.
  3. If I understood, the user, who is concerned about its privacy, is the creator of the blockchain. Then, why do the authors think a user would collude with an agent?
  4. Experimental section is still in poor form. A table of the simulation parameters should be provided. The authors said, “some tasks were not submitted within the specified time”. What is this specified time? What is the impact if this time is increased or decreased?

Reviewer 3 Report

This article propose a location privacy protection method based on blockchain. The proposed method satisfies the principle of k-anonymity privacy protection and does not need the help of a  trusted third-party anonymizing servers.

The paper need to address the following comments

1. Quality of presentation

There are so many typos in the paper that need to be corrected. I will just cite few but the whole paper needs to deep and thorough proofreading

142 Jia et al. [39] Proposed an incentive mechanism using blockchain (the p should not a capital letter)
142 Jia et al. [39] Proposed an incentive mechanism using blockchain
194 1). Requester. Requester is the user who needs to get location service (the dot before Requester is not needed)
198 2). Agent. Agent takes the place of the requester to send query requests
203 3). Miner. The miner is responsible for verifying the service results

2. Result presentation and discussion

The authors discuss few relevant works such as reference 39, 40, 41, and 42 but section 6 which discuss analysis and performance do not deeply diiscuss their results based on those scheme. We suggest a table or figure that summarizes the key security goals achieved by their scheme compared to existing related work. 

Reviewer 4 Report

The authors propose a decentralized location privacy protection scheme based on multiple private blockchains and without the need of a trusted third-party anonymizing server. The main goal is to achieve a balance between the efficiency of queries and the privacy of the users.

Comments

- While listing related works in Section 2, the authors need to clearly position and differentiate their work with respect to them. In addition, the review of privacy-preserving solutions involving blockchain could be further enhanced with state-of-the-art techniques and crypto-solutions. The differentiation with respect to the works in refs. [28] and [41] is also very important for the novelty of the paper.

- Please elaborate on the implications of the proposed method in the case of continuous queries.

- In Algorithm 1, please motivate why each user account ID has to be cleared three times on the private blockchain.

- Please comment on the scalability performance of the proposed scheme for adoption in large-scale scenarios.

- In general, there is a lack of performance comparison with existing approaches. The authors are encouraged to demonstrate quantitative comparison with at least one alternative privacy-preserving method.

There are several typos spotted in the text (examples below). The authors are encouraged to perform proofreading of the entire document.

Page 1, line 43 and Page 2, line 48: “The anonymizing servers is”

Page 2, line 69: “can be two mains categorized”

Figure 3: “Requeser”

Round 2

Reviewer 1 Report

The authors made efforts to address the previous remarks and suggestions about the paper. I think the paper may be accepted at this stage.

Good luck.

Reviewer 2 Report

Thanks to the authors for their effort to improve the paper. Many problems are resolved and I think it is possible to improve it further.

  1. Reward distribution subsection has two different equations for calculating the Rs and Rf. The author should provide some detail on the deduction of those equations or provide proper citations.
  2. Experimental section is still in poor form. A table of the simulation parameters should be provided. The authors said, “No matter what k is, we will fix t = 25s, Tr = 120s.”. Is there any specific reason for keeping the time value fixed?
  3. Please use a proofreading service or ask a professional proofreader to ensure fluency, clarity and coherence without grammar errors.

Reviewer 3 Report

The comments were addressed accordingly. 

English minor check  and thorough proofreading  is required

Reviewer 4 Report

The authors have addressed my concerns raised during the previous submission in a satisfactory manner. In particular, the comparison with the existing privacy protection architectures is now more clear. I recommend acceptance of the paper.

Author Response

This manuscript is a resubmission of an earlier submission. The following is a list of the peer review reports and author responses from that submission.

Round 1

Reviewer 1 Report

Comments to Authors:
In this manuscript, the authors propose to utilize blockchain technology to solve location privacy problem with the third-party anonymizer-based approaches. In order to achieve location privacy, the authors propose to create new blockchain for new queries. The authors simulated the proposed idea on remix blockchain and show some statistics in terms of task success rate, average response time, and response time.

Comments to the authors:

  • The motivation of using blockchain is not clear. The transition from third-party anonymous servers to blockchain is very lousy and not justifiable. There are many decentralized location privacy-preserving approaches. What are their problems?
  • “Anonymous servers” should be “Anonymizing servers”
  • The manuscript claims that, as it is using blockchain, the overall LBS architecture becomes 2-tier. However, it is not. A user still needs to go through an agent to query a LBS. Perhaps, the authors should check the following papers:

Shahid, A.R., Pissinou, N., Njilla, L., Alemany, S., Imteaj, A., Makki, K., & Aguilar, E. (2019). Quantifying location privacy in permissioned blockchain-based internet of things (IoT). MobiQuitous '19.

Li, L., Liu, J., Cheng, L., Qiu, S., Wang, W., Zhang, X., & Zhang, Z. (2018). Creditcoin: A privacy-preserving blockchain-based incentive announcement network for communications of smart vehicles. IEEE Transactions on Intelligent Transportation Systems, 19(7), 2204-2220.

  • Creating new blockchain for every query (to maximize privacy) seems too ambitious in terms of communication and computation cost.
  • Very important question: what are the blockchain nodes?
  • The results are poorly presented. Why “task was unsuccessful” becomes zero for k = 10 in fiure 5 while it was increasing with the value of k?
  • There are many careless mistakes in the paper. For instance, figure 2, “Aownload Task”. In some places, authors are names are capitalized (e.g. SHUHEI et al. [36])

Comments to the editor:

I am not clear with motivation of using blockchain here. The paper gives absolutely no information on the detail of the blockchain. In my view, the paper is forcing the blockchain to solve the problem. The paper is poorly written with many careless mistakes. The papers lack a good analysis of existing works. The creation of new blockchain is too unrealistic.

Reviewer 2 Report

- In this article, the authors propose a blockchain-based approach for preserving  location privacy.

- The plagiarism rate is equal to 21% (according to ithenticate) which is a bit high.

- A brief introduction about  Location-based services (LBS) is missing.  The same applies for the  k-anonymity.

- In the following paragraph:

"In the centralized architecture, users ask a fully-trusted third party to send the query request to the location service provider (LSP). For example, chow et al. [16] firstly proposed a peer-to-peer (P2P) spatial cloaking algorithm for user collaboration."

The authors talk about centralized architectures. However the proposed example is a decentralized one.

-  In the introduction when the authors listed their contributions the used the present tense for the first contributions and the past tense for  the last one.

- In the last paragraph of the introduction, the authors use the roman numbering for describing the structure of the paper. However, they use classical numbering for listing the different sections.

- The subsection dedicated for covering related work about the use of blockchains for preserving location privacy  is too short.

- Moreover, the authors did not identify the limitations of the existing works which could have motivated the proposition of their new approach.

- Two agents appear in Figure 2, however the authors explain the role of only one agent.

- In section 3.3, the authors are supposed to talk about  the adversary model . However no model is proposed. Moreover, the authors mention only to possible attackers without explaining or mentioning the source of this proposition.  (Furthermore, we are not sure whether the authors mention how these possible attacks are avoided by their proposed approach.)

- The tittle of Figure 4 contains a grammatical error. The content of the figure is a bit confusing as well.

- Algorithms 2 and 3 contain a lot of verbose text. Moreover, Algorithm is cut between two pages.

- In section 5.2,  the authors propose a simulation for validating their approach. Thus they  have not made any real experimentation.

- In Figure 5, the meanings of "successful1" and "successful2" are not clear.

- The difference between Figures 6 et 7 is not clear too.

Summary:

- Unfortunately, the paper  is not well written and contains a lot of grammatical and form mistakes.

-  Moreover, the contribution is not significant as well. The authors simply apply existing techniques from the blockchain  field for preserving a location privacy.

- No real experimentation is done.  The proposed approach is validated by simulation.

- In the validation section, the authors concentrated on efficiency/success-rate/response-time and did not consider on privacy and robustness against mentioned attacks which are normally  the main  goals of their work.

- For all these reasons, the authors need to work a bit on improving the quality of the paper and obtained results.